# Maternal Sociodemographic Factors and Antenatal Stress

**DOI:** 10.3390/ijerph18136812

**Published:** 2021-06-25

**Authors:** Maheshwari Andhavarapu, James Orwa, Marleen Temmerman, Joseph Wangira Musana

**Affiliations:** 1Department of Obstetrics & Gynaecology, The Aga Khan University, Nairobi 00100, Kenya; marleen.temmerman@aku.edu (M.T.); Wangira.Musana@aku.edu (J.W.M.); 2Department of Population Health, The Aga Khan University, Nairobi 00100, Kenya; james.orwa@aku.edu

**Keywords:** stress, hair cortisol, cortisone, pregnancy, sociodemographic, income, perceived stress

## Abstract

Antenatal stress has been associated with adverse birth outcomes such as fetal growth restriction, low birth weight, and preterm birth. Understanding key determinants of stress in a vulnerable pregnant population has the potential of informing development of targeted cost-effective interventions to mitigate against these adverse birth outcomes. We conducted a secondary analysis of data from 150 pregnant women attending antenatal care services at a rural referral hospital in Kenya. The participants completed a sociodemographic and clinical questionnaire, the Cohen’s Perceived Stress Scale (PSS) and gave a hair sample for cortisol and cortisone analysis. The association between selected sociodemographic predictors (age, parity, marital status, maternal education, household income, polygyny, and intimate partner violence) and outcomes (hair cortisol, hair cortisone, and PSS score) was examined using univariate, bivariate and multivariate models. We found a negative association between PSS scores and household income (β = −2.40, *p* = 0.016, 95% CI = −4.36, −0.45). There was a positive association of the ratio of hair cortisone to cortisol with Adolescent age group (β = 0.64, *p* = 0.031, 95% CI = 0.06, 1.22), and a negative association with Cohabitation (β = −1.21, *p* = 0.009, 95% CI = −2.11, −0.31). We conclude that household income influenced psychological stress in pregnancy. Adolescence and cohabitation may have an influence on biological stress, but the nature of this effect is unclear.

## 1. Introduction

Stress is a physiological response to the mental, physical, or emotional challenges that we experience in our day-to-day life [1]. Pregnancy is a period characterized by numerous simultaneous psychological and physiological changes and may be a stressful event to the mother [2,3]. However, the cause(s) of stress may vary for each woman, as does their reaction to the stress and their ability to manage or adapt to it.

Several causes of antenatal stress have been identified including the hormone-related symptoms of pregnancy (e.g., nausea, vomiting, constipation, fatigue, pain, lack of sleep, etc.), the emotional effects of the pregnancy (e.g., worry about the labor and delivery process, establishing breast-feeding, the future of the baby, increased family-related responsibility, etc.), work-related stress, environmental factors, e.g., natural disasters [4,5], among others.

Mental stress causes both immediate and long-term changes in the psycho-neuro-endocrine and immunological pathways in the body. This mainly involves the hypothalamus–pituitary–adrenal (HPA) axis, the sympathetic nervous system (SNS) and the limbic system. Under mental stress, the influx of emotions from the limbic system cause hyperactivation of the HPA axis and SNS. Hyperactivation of the hypothalamus results in release of high levels of corticotropin-releasing hormone (CRH) into the hypothalamic paraventricular nucleus. CRH acts on the pituitary gland, stimulating the release of adrenocorticotropin hormone (ACTH). ACTH acts on adrenal glands to stimulate production on glucocorticoids (cortisol) [4]. Cortisol can be metabolized into its inactive form, cortisone, by 11-β-hydroxysteroid dehydrogenase (11B-HSD) type 2. The ratio of cortisone to cortisol is considered an indirect marker of 11B-HSD type 2 activity [6].

Cortisol has several metabolic and physiological effects on the body. In pregnancy, maternal cortisol stimulates the release of CRH by placental cytotrophoblasts, amnion, chorion, and decidual cells. Placental CRH enhances the production of prostaglandins, resulting in rupture of membranes and/or the onset of the process of labor. This is one of the underlying pathophysiological mechanisms for preterm labor [7].

Moderate to severe antenatal stress and anxiety increase the risk of fetal distress, preterm birth (PTB), neonatal crying, and low birth weight (LBW) [4]. Studies have examined and found a relationship between different kinds of stress measures in pregnancy and PTB. Maternal stress has also been linked to cognitive, behavioral, and neurodevelopmental problems in the child in future [8,9]. Long-term consequences to infant, child, and adult health have been seen, with resultant significant health, societal and economic burden globally [10,11].

Antenatal perceived stress has also been associated with post-partum anxiety and depressive symptoms, as summarized in a recent systematic review, resulting in poor maternal psychological health [5].

Overall, the influence of stress on maternal and neonatal health is well known but key determinants of stress in pregnancy are poorly understood, especially in sub-Saharan Africa, where literature is deficient.

### 1.1. Stress as a Concept

Stress is a complex multidimensional entity involving several aspects: the stressor itself (environmental demands); the individual’s resources (social support, socio-economic conditions, lifestyle, and personality); the individual’s perception of stress; and lastly, their adaptation and response to the stress. This makes it challenging to assess both in the pregnant and non-pregnant population and only a few studies assess all the above potential aspects of stress [12]. Maternal sociodemographic characteristics have been studied directly in relation to PTB, but the underlying mechanisms are complex and poorly understood [13,14]. Stress could be the underlying mediating factor via psycho-neuro-endocrinological pathways, but the association of sociodemographic factors (such as maternal education, household income, intimate partner violence, etc.) with maternal antenatal stress has not been explored well in literature. To date, there are no studies in Kenya or even Africa, and only one related study conducted very recently in the United States that examined the association between maternal life-course socioeconomic status and hair cortisol levels (as a biological measure of stress) in pregnancy [15]. Exploring these relationships may improve our understanding of the risk factors of stress in pregnancy and help innovate targeted interventions to mitigate against maternal stress and reduce the burden of adverse birth outcomes.

### 1.2. Measuring Stress in Pregnancy

Pregnancy is a period of profound bio-psycho-behavioral changes and stress is a difficult construct to measure. Assessment of antenatal stress has been made via psychological and physiological measures [16]. Psychometric instruments (validated questionnaires) are commonly used to measure psychological stress in pregnancy, consisting of self-reported information about the respondent’s perception of stress or a stressful event. Several instruments are currently available depending on the construct being assessed, the best ones being the Perceived Stress Scale, Edinburgh Postnatal Depression Scale, Prenatal Distress Questionnaire, Prenatal Life Events Scale, State-Trait Anxiety Inventory, and the Abbreviated Scale for the Assessment of Psychosocial Status in Pregnancy [12]. However, self-reported stress scores can be affected by recall bias, cultural context problems, difficulty in interpretation, trying to provide socially desirable answers, or cognitive reframing, and thus the need for a more objective measure such as a biomarker.

Cortisol is produced as a response to stress and it has been commonly used as a biomarker of stress. However, cortisol levels in serum, saliva, and urine reflect short-term or acute stress and are influenced by several other factors such as feeding, hydration, physical activity, sleeping patterns, and the circadian rhythm [6]. Sampling is also affected by collection methods, participant adherence, and complicated by the need for repeat measurements and special storage conditions [17,18]. This had led to the recent emergence and investigation of a more reliable marker of chronic stress—hair cortisol concentration (HCC) [19]. Mechanisms by which cortisol is incorporated into growing hair follicles are incompletely understood but include passive diffusion from the blood stream, sweat, and sebum secretions, and small amounts of local synthesis directly reflecting HPA axis activity [20]. Rate of hair growth varies with age, race, and sex, and Africans in particular have a lower hair density, slower growth rate, and higher telogen counts compared to Caucasians [21] on which most hair cortisol research has been based. A growth rate of 1 cm/month is an accepted average [22], and thus a 3 cm hair segment closest to the scalp would reflect the retrospective accumulation of cortisol over the past 3 months (chronic stress). However, cortisol concentrations tend to decline distally from the scalp as a result of “wash-out” and damage from sunlight, washing and cosmetics, and hence this limits the retrospective period that can be analyzed using hair cortisol concentrations [23]. Hair growth also varies among scalp regions, the posterior vertex having the most uniform growth rate, lowest proportion of follicles in resting phase and least inter-individual variation [15], thus making it the standard region for hair collection. Sampling technique is easy and non-invasive, and storage is at room temperature [24]. Hair cortisol has been shown to correlate well with stress in pregnancy [6,25,26,27] and rise throughout pregnancy especially in the third trimester [27]. Its metabolite, hair cortisone, is still under evaluation as a biomarker of stress based on scant studies.

### 1.3. Sociodemographic Factors and Psychological (Perceived) Stress

The impact of a stressor on the health of an individual may be determined by the individual’s perception (and hence, their reaction) of it and not necessarily the stressor directly. Individuals actively interact with their environments and the effects of stress are thought to occur only when both (a) the situation is adjudged by the individual as threatening or demanding, and (b) the resources available to cope with the situation are insufficient, implying that the effect is not merely based on the intensity or inherent quality of the stressor or stressful event but rather is dependent on personal and contextual factors as well [28].

The Perceived Stress Scale (PSS) is one such instrument that was developed by Cohen, Kamarck, and Mermelstein in 1983, taking the above theory into account. It is an accepted near-universal psychometric tool for measuring perceived stress and has been validated widely in different languages and cultural contexts [29,30,31,32,33,34,35]. It consists of either 4, 10, or 14 items (3 versions) on a five-point Likert scale that capture the global perception of stressors by the respondent. It measures the degree to which situations are considered unpredictable, uncontrollable, and burdensome by the respondent during the past month. It can also be used as an outcome variable [36]. As demonstrated by a systematic review in 2013, it adequately measures mental disturbances by daily hassles and has been identified as the best instrument currently available in this category for its excellent reliability data in pregnant samples and its high validity data in non-pregnant samples [12]. After 4–8 weeks, however, its predictive validity may fall as perceived stress is dynamic and is affected by changes in coping [36].

Sociodemographic associations of PSS have barely been studied. Analysis of data from three national surveys in the United States, and normative data from a German sample revealed increasing psychological stress (PSS scores) with decreasing age, income, and education and higher scores among women, unmarried, and unemployed individuals [37,38]. However, others found PSS scores not to be affected by marital status [39,40].

Parity has barely been explored in relation to perceived stress and the few existing publications are contradictory demonstrating PSS scores as either lower or comparable in primigravidae compared to multigravidae [41,42].

A cross-sectional study by Kashanian et al. in 2019 looking at perceived stress and stressors in 200 immediate post-partum women revealed higher PSS scores in women exposed to their partner’s verbal and physical aggression [43]. Several other studies have shown higher perceived stress [44], anxiety, depression, and post-traumatic stress disorder among women exposed to partner violence [45,46,47,48,49], very few of which included pregnant women. We found no literature on perceived stress in relation to polygyny.

### 1.4. Sociodemographic Factors and Biological Stress

Some determinants of HCC have already been explored in a few studies in various settings, mostly in the developed world where chronic stressors may vary from those in the developing world and low-middle income countries [50].

A meta-analysis in 2017 on the stress-related and basic determinants of hair cortisol revealed its positive association with age, male sex, less frequent hair washing, body mass index, waist-to-hip ratio, and systolic blood pressure, and no consistent associations with self-report measures of perceived stress. Of the 66 studies included in the meta-analysis, only two involved a sample of pregnant women and both these studies aimed at exploring the utility of hair cortisol as a biomarker of stress, but did not further investigate determinants of HCC [51].

With regards to sociodemographic factors, to date, only a single study has interrogated hair cortisol in relation to maternal life-course (childhood and pregnancy) socioeconomic status (maternal education, annual family income, and house ownership) revealing a negative association [15]. In Kenya, there has been only one study assessing hair cortisol among settlement communities in Naivasha and found increased levels among females, divorced individuals, and those earning below-minimum wages. The hair cortisol concentrations were also higher overall as compared to a Caucasian reference group [52].

A meta-analysis published in 2019 [53] assessed the strength and direction (increase or decrease) of association between adverse life events such as maltreatment, natural disasters, accidents, domestic violence, etc., and hair cortisol, concluding that there is small to moderately significant association between the two and the strength and direction was dependent on the timing and type of the adverse event. Only three of these studies involved women subjected to domestic violence, two of which demonstrated higher hair cortisol levels in women who experienced intimate partner violence [54,55]. None of the studies included in this meta-analysis assessed partner violence during pregnancy.

Parity and polygyny as determinants of hair cortisol have not been explored in literature.

Hair cortisone concentration (HCNC) has been found to be positively associated with psychological stress [56], and with age, male sex, and diabetes mellitus (just as hair cortisol) [57]. Staufenbiel et al., in his study looking at determinants of cortisol and cortisone, found no influence of educational level on cortisone or cortisol concentrations [57]. In pregnancy, one study assessed changes in hair cortisone and cortisone/cortisol ratio and found both, but not cortisol alone, to be associated with stress in the second and third trimester of pregnancy, and hence suggesting that both cortisone and cortisol should be used as a marker of stress in pregnancy [6]. Other determinants of HCNC, including sociodemographic, are unknown.

Limited studies have looked at the sum and ratio of HCC and HCNC. Given that cortisol may be converted to cortisone in hair, measurement of total hair glucocorticoid may provide a more accurate representation of systemic cortisol levels and HPA axis activity. However, research involving total glucocorticoid in hair so far have shown no relation to stress and a positive association with age (as for cortisol and cortisone) [57,58]. No determinants and associations of the ratio of cortisone to cortisol have been established so far.

Table 1 below summarizes current available literature on selected sociodemographic determinants of psychological (perceived stress using PSS scores) and biological stress measures (hair cortisol and cortisone), including the non-pregnant population.

The two types of stress measures, biological and psychological, do not correlate well and this has been shown in a few studies, including one where HCC was higher with higher perceived stress levels but lower with highest perceived stress [55,59,60]; hence, the need for both measures when assessing stress as a whole.

### 1.5. The Gap in Literature

As shown in Table 1 above, to the best of our knowledge, the influence of sociodemographic factors on both biological and psychological stress measures is largely unknown, more so, in pregnancy despite the known effects of stress on maternal and neonatal health.

Kenya is a low-middle income country (LMIC) in Africa with a high prevalence of PTB (12%) and where pregnant women are exposed to multiple chronic life stressors [52,61] in terms of socio-economic conditions. Most data exist in the Western population, which may differ from our population in terms of socio-economic status, exposure to stressors, coping strategies, and resources. Race/ethnicity have also been shown to affect HCC [62] in some studies and this is thought to be due to differences in lifestyle, genetics, sociocultural, and possible environmental factors [63]. In addition, findings on socioeconomic determinants from publications so far are inconsistent and based on small studies.

The primary objective of this study was to determine the association of selected maternal sociodemographic factors with psychological and biological stress in pregnancy. The secondary objective was to further explore the differential effects of the selected maternal sociodemographic factors on the sums and ratios of hair cortisone and cortisol concentrations. Some factors may influence stress in more profound ways than others, taking into consideration key confounding factors, and this information will enable targeting of interventions to reduce the effects of stress on mother and child [61,64].

## 2. Materials and Methods

### 2.1. Study Design and Population

This was a secondary analysis of data from a cohort study which was conducted among 150 pregnant women between 22- and 28-weeks’ gestation attending antenatal clinical care services at Migori County Referral Hospital in Kenya. The objective of the parent study was to examine the differential influence of perceived stress versus hair cortisol concentration during pregnancy in predicting gestational length among pregnant Kenyan women [65].

Migori County is in south-western Kenya, bordering Tanzania, Uganda, and Lake Victoria. It is approximately 368 km from Nairobi—the capital of Kenya—with a diverse population of about 1,126,000 including Kenyans, Indians, Somalis, and Arabs. The population density is approximately 435 people per sq. km, with a poverty rate of 32% and an unemployment rate of above 8%. Their main economic activities include fishing, manufacturing (especially sugar), agriculture, and goldmining. There are three main languages spoken by the inhabitants—Dholuo (local language), Kiswahili (national language), and English. There is only one main road of bitumen standard which is the highway to Tanzania, the rest are mostly earth roads, and some, gravel. In addition to the county referral hospital, there are four district hospitals, five sub-district hospitals, one catholic mission hospital, and several health centers. Although there are about 180 secondary schools, only 15% of the residents have a secondary level of education or above. There are very few tertiary training institutions and only approximately 2% of the population have tertiary or university level education [66].

A sample size of 150 for the parent study was based on a regression model that considered inclusion of 2 independent variables and 2 covariates with an effect size set at 0.10, desired power of 0.8, probability of 0.05, and an attrition rate of 20%. Inclusion criteria was women aged 18–45 years with a singleton pregnancy between 22- and 28-weeks’ gestation who were interested in participating in the study, able to speak and understand at least one of three languages (Dholuo, Kiswahili, or English) and planned to deliver at the Migori County and Referral Hospital. Exclusion criteria consisted of women with the following obstetric, medical, or health complications—multiple gestation, fetus known to have structural or genetic abnormalities, known uterine or cervical anomalies, history of placenta previa or abruption, chronic renal or heart disease, diabetes mellitus, hypertension, thyroid disease, adrenal disease, HIV infection or AIDS, cognitive impairment, and mental health disorders; those whose pregnancy was achieved by artificial techniques and those who had bleached their hair with peroxides or other chemicals known to affect hair cortisol measurement. All eligible participants completed a sociodemographic and clinical questionnaire (Appendix A), a Perceived Stress Scale (PSS-10, Appendix A), gave a 3 cm long hair sample from the posterior vertex of the scalp for cortisol and cortisone analysis antenatally, and after delivery, completed the Perinatal Medical Risk Index (PMRI) form (Appendix A). Details on laboratory processing of hair samples are available in the parent study’s publication [65].

Retrieved from the sociodemographic questionnaire, the selected independent (predictor) variables in this study included age, parity, marital status, maternal education, household income, polygyny, and intimate partner violence. Among the last questions in the tool, women were asked whether they had been subjected to any violent behavior from their partners since becoming pregnant, and those who stated “yes” were required to choose the type of violence from a list of options or specify others (No. 36, Appendix A). Age was collected as a continuous variable and parity as an ordinal variable. Marital status, maternal education, household income, polygyny, and intimate partner violence were categorical variables. The primary dependent (outcome) variables included hair cortisol concentration (HCC), hair cortisone concentration (HCNC), and PSS score. The secondary outcome variables were the sum of hair cortisone and cortisol, and the ratio of hair cortisone to hair cortisol. All outcomes were continuous variables. The parent study employed the PSS-10 which consists of 10 items on a 5-point Likert scale (where 0 = “never”, 1 = “almost never”, 2 = “sometimes”, 3 = “fairly often”, and 4 = “very often”) from which the respondent chooses appropriately. PSS scores are calculated by reversing the scores of the positively-stated items (items 4, 5, 7, and 8) such that 0 = 4, 1 = 3, 2 = 2, 3 = 1, 4 = 0, and then summing scores across all items. Higher scores mean higher perceived stress and there is no strict cut-off but generally, scores ranging 0 to 13 are considered mild or low stress, 14 to 26 as moderate, and 27 to 40 as high perceived stress.

### 2.2. Data Management and Analysis

Stata 16 was used for statistical analysis. For descriptive analysis (*n* = 150), each of the continuous predictor and outcome variables were explored for normality using the Shapiro–Wilk Normality Test and expressed as mean (standard deviation, SD) or median (interquartile range, IQR) for normal and non-normal distributions, respectively. Categorical data were expressed as frequencies and percentages. HCC and HCNC were represented in picograms per milligram (pg/mg) of hair. This was converted to femtomoles (fmol/mg of hair) for calculation of the sums and ratios of cortisone and cortisol.

Categorical variables with categories that had too few observations or that would have no meaningful interpretation after analysis were collapsed (household income and marital status, respectively). Maternal education had a category “none” containing only 1 observation and household income had a category for 1 missing observation, and these were dropped to prevent interference during bivariate analysis (*n* = 148). Data were then explored for validity of linear regression by examining whether its assumptions held—continuous predictors were explored for collinearity using the Pearson’s Bivariate Correlation test and a correlation matrix was obtained, and non-normally distributed outcome data were log-transformed (HCC, HCNC, sums and ratios of cortisol and cortisone). Explorative bivariate analyses using simple linear regression were then conducted to examine the association between each predictor variable and each of the 5 outcomes (3 primary and 2 secondary). Age was explored both as a continuous and a categorical variable to look for any age-group-specific trends.

The influence of known and possible confounders—PMRI, baby’s gender, and body mass index (BMI)—was also assessed in the bivariate model. Enlow et al., in their study on maternal cortisol output, found higher cortisol levels in all three trimesters among mothers of male neonates compared to those of female neonates [67]. BMI has been found to be a covariate of cortisol in a meta-analysis [51]. The PMRI recorded any current or past obstetric conditions (a potential source of maternal stress) represented as an overall score.

Missing data were then excluded resulting in a complete data set of 110 observations per variable (*n* = 110) for the multivariable linear regression. Eligibility for entry into a multivariate model was a *p*-value < 0.25 in the bivariate model based on the work on linear regression by Bendel et al. and Afifi et al. [68] (p. 95). Eligible predictors were simultaneously entered into a multivariate model (mutual adjustment) to determine their independence in portending PSS, HCC, and HCNC at a significance level of 0.05.

## 3. Results

### 3.1. Descriptive Data

The sociodemographic characteristics of the study participants are summarized in Table 2 below. The median age was 24 years (IQR: 17–35 years). A large proportion of pregnant women were adolescents (35.81%) and almost an equivalent number (37.84%) were of advanced maternal age (>35 years). Almost a third (36.91%) were nulliparous women. The median BMI was 25.7 kg/m^2^ (IQR: 19.9–38.5 kg/m^2^), just into the overweight category. About 8% of the women were single, two-thirds were married, and none were divorced or separated. The monthly income of majority of the women were either nil or less than 5000 Kenyan shillings, while approximately 10% earned above 20,000 Kenyan shillings. Majority (71%) had attained at least secondary level of education. Forty-two percent were unemployed. Two-thirds had a monthly household income below 10,000 Kenyan shillings. Polygyny was reported by 25 women (17.61%) and intimate partner violence by 27 women (18.24%). About 12% of the study participants ended up having a preterm delivery in the current pregnancy.

The Perceived Stress Scale (PSS) scores were normally distributed with a mean (SD) of 19.24 (4.05). Both hair cortisol and cortisone levels were positively skewed with medians of 14.7 pg/mg (1.5–12.5 pg/mg) and 10.06 pg/mg (5.0–12.0 pg/mg), respectively. The sums and ratios of cortisone and cortisol were also positively skewed with median (IQR) of 68.63 fmol/mg (21.77–70.56 fmol/mg) and 1.007 (0.797–1.014), respectively.

### 3.2. Bivariate Analysis

Table 3 below summarizes the findings of bivariate linear regression analysis of each of the five (three primary and two secondary) outcome variables against the seven selected predictor variables. Detailed tables including the respective β estimates and 95% confidence intervals (CI) are attached as Appendix A. All outcomes had predictor variables with *p*-values below 0.25 except HCNC which had only one such variable—Prenatal Medical Risk Index (*p* = 0.11)—and hence a multivariate model was not run for HCNC as an outcome.

### 3.3. Multivariate Analysis

Table 4 below summarizes results of multivariate regression analysis for PSS scores, HCC and the secondary outcomes against the eligible predictors. Detailed tables including the respective β estimates and 95% CI are attached as Appendix B. A monthly income of above 20,000 Kenyan shillings had a significant negative association with PSS scores (*p* = 0.016, β = −2.405, 95% CI = −4.356, −0.453). There were no associations found of statistical significance between any of the predictors and either HCC or total glucocorticoids (Appendix B, Table A6 and Table A7). However, the adolescent age group showed a significant positive association with the ratio of cortisone to cortisol (*p* = 0.031, β = 0.636, 95% CI = 0.058, 1.215) and “Cohabitation” as marital status had a significant negative association with the ratio (*p* = 0.009, β = −1.206, 95% CI = −2.107, −0.306).

## 4. Discussion

Our study is among the first of its kind in Africa and indeed globally to report on maternal sociodemographic determinants of stress whose in-depth understanding can help device simple targeted interventions aimed at reducing maternal stress especially in low to middle income countries where the burden of adverse birth outcomes is high.

### 4.1. Age and Stress Measures

Most literature show a positive association of age with hair cortisol, hair cortisone, and their sum, possibly due to the mild age-related hypercortisolism because of reduced HPA axis sensitivity. However, our results contradicted this, possibly due to restricted age variance in our sample or the small sample size.

The cortisone to cortisol ratio, on the other hand, was higher among the adolescent age-group. This age-group has barely been studied in literature in relation to stress and hence stress reactivity is largely unknown. The ratio is thought to be an indirect marker of local activity of the enzyme, 11-β-hydroxysteroid dehydrogenase, which converts cortisol to cortisone. Little is known about this enzyme and it may be susceptible to genetic and other unknown factors. Higher ratios mean higher cortisone or lower cortisol concentrations and the growth spurt in adolescents may explain higher cortisone from conversion of higher circulating cortisol levels. It is possible that the association we found in our study between adolescent age group and higher cortisol/cortisone ratio reflects normal biological mechanisms, but this association needs further exploration.

Age was not associated with PSS scores unlike the findings of Cohen et al. in a U.S. population and Klein et al. in a German population which showed a negative association.

### 4.2. Marital Status and Stress Measures

Whether married, single, or divorced was not associated with hair cortisol or cortisone, and we found no comparative studies in literature. However, Chin et al. found lower salivary cortisol levels in a sample of 572 married men and women, compared to never or previously married individuals. We chose marital status as a predictor of stress because of the possibility of single mothers having higher levels of stress as caring for a new-born alone may be more emotionally, physically, and financially challenging, as compared to mothers with partners. Notably, in the present study, single women accounted for only 8% (12 women) and this may have affected the results. In addition, single women may have help and support readily available from other family members in anticipation.

There was no association with PSS scores, as supported by Chin et al. who found no difference in average PSS scores in a study investigating marital status as a predictor of perceived stress and salivary cortisol. However, findings from a study by Cohen et al. on 2387 respondents, looking at variation of PSS scores among sociodemographic factors found that married individuals scored lower than single or divorced individuals. Married people tend to be healthier and less stressed but the exact mechanism underlying this buffer effect is unknown.

Cohabitation appeared to be negatively associated with the ratio of cortisone to cortisol. This is an unexplored area in literature and provides opportunity for deeper research as to whether cohabitation could be a protective factor and the underlying mechanism.

### 4.3. Polygyny and Stress Measures

There was no significant association between polygyny and either biological or psychological stress measures. We had expected to find higher levels of stress in pregnant women whose partners had a polygynous relationship. The study population had a relatively high prevalence of polygyny (17.61%) but also mostly consisted of the Luo tribe among whom polygyny is relatively common and probably also better accepted by women, and this may explain the lack of influence on stress. To the best of our knowledge, we found no literature looking at the influence of polygyny on stress, hence making these index findings.

### 4.4. Household Income and Stress Measures

A monthly household income of above 20,000 Kenyan shillings appeared to reduce PSS scores in pregnancy. Data on PSS scores of a U.S. probability sample in 1988 by Cohen et al. revealed a decrease in PSS scores with increasing household income. A 2016 social sciences randomized control trial by Shapiro et al. on the effect of cash transfers on perceived stress on a Kenyan population revealed improved mental well-being as evidenced by improvement in the four-item PSS scores. No associations were found between household income and the biomarkers (HCC, HCNC, sums and ratios of cortisone and cortisol), contradictory to the findings of Enlow et al. in the only study of socioeconomic status of pregnant women, and Henley et al. on Kenyan settlement communities. Another longitudinal study by Serwinski et al. also revealed a negative association of hair cortisol with income in 164 middle-aged women.

Several reasons may explain increased stress due to low household income in pregnancy including the increase in expenses to attend antenatal care (transport, and time at clinics is lost work hours), medications in pregnancy, care of the expected new-born and possibly other younger children, and reduced productivity or work hours due to pregnancy-induced physiological effects/changes on the body.

Out of pregnancy, higher mortality rates have been seen among those with larger losses in earning as a result of increased stress [69].

Some research in Kenya on the impact of financial benefits such as unconditional cash receipts and health insurance on stress has revealed lower perceived stress and cortisol levels [70,71], health insurance being better at reducing stress via a “peace of mind” effect [72,73]. Interventions to improve and/or protect household incomes, such as the provision of health insurance with subsidized affordable premiums, can mitigate against maternal stress.

### 4.5. Maternal Education and Stress Measures

We expected maternal education to reduce biomarkers of stress and/or PSS scores, as shown by Enlow et al. and Cohen et al. but our results showed no such association. Higher education may reduce stress levels through various mechanisms including proper employment opportunities and income, differences in perception of specific events as stressful, better management of stress including coping strategies, better availability of resources to cope effectively, and lastly, knowledge of and/or accessibility to psychological support. In our sample, possibly the influence of education is masked by other factors such as cultural practices that may affect behavior/approach to stressful events and the availability of social support in pregnancy.

### 4.6. Parity and Stress Measures

There was no association of parity with either biological or psychological measures of stress. Lack of association with perceived stress was consistent with those of Wheeler et al. who found no difference in PSS scores of primiparous compared to multiparous women in a study of 1606 pregnant women. With regards to biomarkers, we found no existing literature, making these index findings. We expected higher stress levels with parity possibly due to anxiety from previous obstetric events, and the anticipated increase in responsibilities, expenses, and the additional time and effort invested in caring for yet another child.

### 4.7. Intimate Partner Violence and Stress Measures

Intimate partner violence (IPV) was reported by 18.24% of women in this study. As per the Kenya Demographic and Health Survey (KDHS) 2014 data, although data specific to IPV in pregnancy was not available, 9% of women experienced physical violence during pregnancy (any perpetuator), and 47.1% of women experience either physical, sexual, or emotional spousal violence unrelated to pregnancy. A 2017 mixed method cross-sectional study of IPV in 238 pregnant women in West Pokot county in Kenya revealed a much higher prevalence 66.9% of overall IPV in pregnancy, significantly associated with partner’s level of education and alcohol intake [74]. Similarly, Makayoto et al. found a 37% prevalence of at least one form of IPV among 300 pregnant women in a cross-sectional study in Kisumu, Kenya [75]. Again, having a partner who had attained tertiary education was protective. The lower prevalence of IPV in our study population may be attributed to a higher proportion of partners (42.3%) having a tertiary level of education acting as a protective factor, or that almost a third of the women themselves had attained a tertiary education compared to the population average of 2%. Additionally, the list of options under IPV in the data collection tool might not have been exhaustive. Although in-depth investigation into IPV and its associated factors was beyond the scope of this study, it is an area worth investigating further due to its known negative impact on pregnancy outcomes.

IPV showed a significant positive association with PSS scores in bivariate analysis that ceased to exist upon multivariate analysis with mutual adjustment for other factors, possibly because of our small sample size with 27 women reporting IPV in pregnancy. Our findings were contradictory to those of Kashanian et al. in a sample of 200 immediate post-partum women who had significantly higher PSS scores when exposed to physical and verbal spousal violence. However, these are important findings as there are no comparative studies in literature exploring IPV and perceived stress in sub-Saharan Africa, although other constructs of mental health such as anxiety and depression have been explored. Boeckel et al. also demonstrated higher hair cortisol levels among women who experienced IPV (*n* = 27) compared to the control group (*n* = 25).

Cultural factors may play a role in mediating the effect of partner violence on mental health as this may affect the perception process; that is, if the mother does not perceive spousal physical, emotional, or verbal aggression as stressful, then her mental health would not get affected, and hence stress measures remain unaltered. Another explanation for lack of association may be the time frame; that is, the PSS measures perceived stress over the past one month beyond which the validity is affected by coping strategies that may change PSS scores, and a 3 cm hair sample represents chronic stress over the past three months. If a mother did not experience any violence around the time of data collection, stress measures may remain unaffected. Another factor that affects stress experienced by the mother is the severity and frequency of the violence and whether the violence had an underlying intention to harm her. Moreover, the PSS may not exactly be the best measure of interpersonal stress and has not been validated for the same. Overall, qualitative research is needed in our setting to understand the effect of partner violence on maternal stress, including the nature of the violence and coping mechanisms.

Two systematic reviews and meta-analyses in 2016 [76,77] have shown a positive association of IPV in pregnancy with preterm birth (PTB) and low birth weight (LBW), and an additional cross-sectional Zimbabwean study in 2018 showed specifically emotional IPV in pregnancy to be positively associated with PTB and LBW [78]. It is possible that cortisol via the HPA axis is the mediating factor, as shown by a meta-analysis by Khoury et al., where the long-term impact of adversity (several types including domestic violence) on dysregulation of the HPA axis was reflected by higher hair cortisol levels. Comprehensive active screening for IPV in pregnancy during antenatal visits and provision of adequate support may mitigate the effects of IPV on stress and its consequences on mother and baby.

### 4.8. Limitations and Strengths

The cross-sectional nature of this study prevents causal inferences from being made. Secondly, some of the assumptions of linear regression were not met and had to be relaxed for the analysis. Lastly, the parent cohort study was not powered to investigate the associations between sociodemographic factors and PSS score, hair cortisol and cortisone concentrations or their sums and ratios. Some associations in this study were significant on bivariate and not on multivariate analysis and some of the findings were contradictory to much larger studies and we attribute this to a possible small sample size and the nature of the study design. In addition, a recognized limitation of self-reported measures such as PSS scores is the possibility of bias due to recall problems or trying to provide socially-desirable answers.

This study had a few strengths, in that, it was the first study to look at several sociodemographic factors in relation to pregnancy in Kenya and even sub-Saharan Africa. Secondly, it involved analysis of both biological and psychological stress measures which may not always be correlated, as seen in the results. Lastly, we utilized samples from a parent study to answer the research question; hair samples were processed abroad in an international laboratory and this is a costly process, hence reducing cost of this study.

## 5. Conclusions

Key maternal sociodemographic factors had modest or no impact on both psychological and biological stress measures. Maternal income influenced the psychological stress measure more than biological stress measures. Adolescent age and cohabitation appeared to have an influence on the biological measures (ratio of cortisone to cortisol). The meaning of this latter finding needs further exploration especially on the impact on adolescent pregnant women and women in cohabiting unions.

Unlike associations demonstrated by literature, we found none with regards to hair cortisol and cortisone. It is noteworthy that climatic conditions and hair growth in Sub-Saharan Africa differ from other populations and studies have shown a possible “wash-out” effect on hair cortisol concentrations by sunlight.

Slower hair growth rates mean that a certain length of hair represents a longer period of cortisol exposure and if that period included a stress-free time, there would be a resultant “dilutional effect” on the increased cortisol secreted in response to stressors.

Interventions to protect or improve household income, such as financial health benefits, health insurance, etc., might be beneficial in alleviating maternal psychological stress in pregnancy. Further research is imperative on the relationship between hair cortisol, cortisone and their ratios and age, especially on pregnant adolescent mothers, to yield a biological explanation of stress vulnerability, and hence special interventions for this group. Determination of whether hair cortisol and cortisone are potential biomarkers of chronic stress, other key determinants of stress among pregnant women and the presence and influence of any resilience factors in sub-Saharan Africa warrants investigation as findings from this study contradicted known predictors of stress.

## Figures and Tables

**Table 1 ijerph-18-06812-t001:** Existing associations of sociodemographic factors with psychological and biological measures.

Predictor	Outcome		
	PSS Score	HCC	HCNC
Age	⇩ [37,38]	⇧ p [51]	⇧ [57]
Parity	⇕ p [41,42]	?	?
Marital Status	⇕ [37,38,39,40]	⇩ * [52]	?
Income	⇩ [37,38]	⇩ p [15,52]	?
Education	⇩ [37,38]	⇩ p [15]	⇔ [57]
Polygyny	?	?	?
Intimate Partner Violence	⇧ p [43,44]	⇧ [54,55]	?

Note: PSS = perceived stress scale, HCC = hair cortisol concentration, HCNC = hair cortisone concentration, “⇕” = mixed findings, “⇧” = positive association, “⇩” = negative association, “⇔“ = no association, “?” = unexplored associations, and “p” = includes studies involving pregnant women. * Married individuals have lower hair cortisol levels.

**Table 2 ijerph-18-06812-t002:** Socio-demographic characteristics of study participants.

Characteristics ^1^	Categories	*n* ^2^	% ^2^
Age (years) (*n* = 148)	≤19 (Adolescents)	53	35.81
	20–34	39	26.35
	≥35 (Advanced maternal age)	56	37.84
Parity (*n* = 149)	Primiparous	55	36.91
	Multiparous	94	63.09
Marital Status (*n* = 148)	Single	12	8.11
	Cohabitation	41	27.70
	Traditional Marriage	71	47.97
	Civil Marriage	8	5.41
	Religious Marriage	16	10.81
Maternal Income (KES ^3^) (*n* = 135)	No income	24	17.77
	≤5000	58	42.96
	5,001–10,000	23	17.03
	10,001–20,000	14	10.37
	20,001–30,000	9	6.67
	30,001–40,000	3	2.22
	40,001–50,000	2	1.48
	>50,000	2	1.48
Household Income (KES) (*n* = 135)	≤10,000	56	41.48
	10,001–20,000	32	23.70
	20,001–30,000	13	9.63
	30,001–40,000	9	6.67
	40,001–50,000	9	6.67
	>50,000	16	11.85
Maternal Education (*n* = 149)	None	1	0.67
	Primary	42	28.19
	Secondary	60	40.27
	Tertiary	46	30.87
Maternal Occupation (*n* = 150)	Housewife/Unemployed	63	42.00
	Self-employed	37	24.67
	Waged Employee	50	33.33
Polygynous Relationship (*n* = 142)	Yes	25	17.61
	No	117	82.39
Current Pregnancy Outcome (*n* = 141)	Term	124	87.94
	Preterm	17	12.06
Baby’s Sex (*n* = 140)	Male	65	46.43
	Female	75	53.37
Intimate Partner Violence (*n* = 148)	Yes	27	18.24
	No	121	81.76

^1^ most variables had some missing observations; ^2^ data presented in frequency and percentages; ^3^ Kenyan Shilling.

**Table 3 ijerph-18-06812-t003:** Summary of bivariate analysis using simple linear regression.

Predictors	*p*-value				
	logHCC	logHCNC	logSum ^1^	logRatio ^2^	PSS Score
Age (years)	0.97	0.62	0.34	0.71	0.99
≤19 (Adolescents)	**0.09 ***	0.51	0.23	**0.14**	0.61
20–34	ref ^3^	ref	ref	ref	ref
≥35 (Advanced)	0.29	0.47	**0.16**	0.56	0.89
Parity	**0.21**	0.54	0.55	**0.06**	0.52
Marital Status					
Single	ref	ref	ref	ref	ref
Cohabitation	0.34	0.50	0.69	**0.06**	0.85
Married	0.92	0.52	0.30	0.57	0.38
Household Income (KES)					
≤10,000	ref	ref	ref	ref	ref
10,001–20,000	0.66	0.47	0.73	**0.18**	**0.22**
>20,000	0.54	0.84	0.60	0.25	**0.02**
Maternal Education					
Primary	ref	ref	ref	ref	ref
Secondary	**0.03**	0.98	**0.18**	**0.02**	0.96
Tertiary	0.60	0.26	0.68	**0.16**	**0.07**
Polygyny	**0.08**	0.69	**0.14**	**0.02**	0.95
Intimate Partner Violence	0.62	0.98	0.97	0.70	**0.02**
Body Mass Index	0.54	0.66	0.71	0.89	**0.08**
Prenatal Medical Risk Index	0.78	**0.11**	0.98	**0.12**	**0.08**
Male baby	0.79	0.48	0.41	0.94	0.44

^1^ log of the sum of hair cortisol and cortisone; ^2^ log of the ratio of hair cortisone to cortisol; ^3^ the category used as the base group in regression analysis. * Variables with *p*-values in bold (*p* < 0.25) were included in the multivariable analysis.

**Table 4 ijerph-18-06812-t004:** Summary of multivariate analysis using simple linear regression.

Predictors	*p*-Value			
	PSS Score	logHCC	logSum	logRatio
Age (years)	- ^1^			
≤19 (Adolescents)		0.051	0.249	**0.031**
20–34		ref	ref	ref
≥35 (Advanced)		0.813	0.577	0.439
Parity	-	0.323	-	0.236
Marital Status				
Single	-	-	-	ref
Cohabitation				**0.009**
Married				0.313
Household Income (KES)				
≤10,000	ref	-	-	ref
10,001–20,000	0.210			0.313
>20,000	**0.016 ***			0.638
Maternal Education				
Primary	ref	ref	ref	ref
Secondary	0.191	0.245	0.414	0.148
Tertiary	0.937	0.966	0.948	0.691
Polygyny	-	0.311	0.333	0.071
Intimate Partner Violence	0.754	-	-	-
Body Mass Index	0.655	-	-	-
Prenatal Medical Risk Index	0.106	-	-	0.958
Male baby	-	-	-	-

^1^ these variables were not included in their respective multivariate models. * values in bold were statistically significant at *p* < 0.05.

## Data Availability

The data presented in this study are available on request from the corresponding author. The data are not publicly available due to privacy as it is still being used for analysis by investigators from the parent cohort study and multiple institutional permissions are required to avail data publicly. Once all analysis on the data has been conducted, the data will be availed in a publicly accessible repository.

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
