# Peer review of "Maternal Sociodemographic Factors and Antenatal Stress"

_ijerph, 2021, doi:10.3390/ijerph18136812_

Round 1
Reviewer 1 Report
In this Research Article, the authors studied the association of perceived stress scale, hair cortisone/cortisol and sociodemographic predictors among pregnant women using univariate, bivariate and multivariate models. Overall, the manuscript is well referenced to justify their scientific rigor. The statistical plans were clearly described, and results were carefully discussed. Most related studies were done in the US. Therefore, the study addressed the gap in knowledge among pregnant women especially in a low-middle income country in Africa with a high prevalence of pre-term birth. A few questions I would like the authors to further address:
- Section 1.4: If the authors can highlight studies on hair cortisol and hair cordisone, it will be clearer to the readers.
- Table 3 presented the summary of all outcomes in the bivariate analysis using simple linear regression, while Table 4-7 presented complete results of multivariate regression analysis for PSS scores (Table 4), HCC (Table 5), sum of HCC and HCNC (Table 6), the ratio of cortisone to cortisol (Table 7). I feel Table 4-7 can also summarized to a table like Table 3 does. Also, why no HCNC were analyzed by itself?
Author Response
Response to Reviewer 1 Comments
Point 1: Section 1.4: If the authors can highlight studies on hair cortisol and hair cordisone, it will be clearer to the readers.
Response 1: These have been well-referenced under section 1.4 and are exhaustive. Studies on use of these as biomarkers for stress are also highlighted under section 1.2 lines 119-121.
Point 2: Table 3 presented the summary of all outcomes in the bivariate analysis using simple linear regression, while Table 4-7 presented complete results of multivariate regression analysis for PSS scores (Table 4), HCC (Table 5), sum of HCC and HCNC (Table 6), the ratio of cortisone to cortisol (Table 7). I feel Table 4-7 can also summarized to a table like Table 3 does.
Response 2: Table 4 now presents a summary of results of multivariate analysis, and detailed results are presented under Appendix B as Tables A6-A9. Please see marked up changes in the attached revised manuscript.
Point 3: Also, why no HCNC were analyzed by itself?
Response 3: Upon bivariate analysis, only one predictor (prenatal medical risk index, p = 0.11) had a p<0.25 (criteria for entry into multivariate model) and hence a multivariate analysis was not conducted. In addition, this was not one of the predictor variables of interest, but instead treated as potential confounder in this study, with a p-value >0.05 and hence considered non-significant.
Reviewer 2 Report
The current manuscript examines the associations between some sociodemographic factors and stress in pregnant women. The manuscript has notable strengths, including the very important and - interestingly highlighted – topic; an attempt to fill the gap in the literature concerning psychological and physical health of women in Sub-Saharan Africa; also, a well-elaborated statistical analysis of collected data; and clear language, which I (not being an expert in medical/biological science) appreciate highly.
Surprisingly, despite its potentials (and despite the evidence found in relevant literature), the study brings just one interpretable result (that pertaining to a link between household income and psychological stress), and two more findings (concerning adolescence and cohabitation) as factors of unclear influence on stress. Although the Authors made a great effort to discuss the results scrupulously, I would like to indicate some other problematic issues related mainly to methodological aspects of the study.
My main concern is about the measurement of outcome variables. In the study, two kinds of stress measures were used: biomarkers as objective indicators of stress and perception of stress as a subjective self-reported measure. I am not in a position to question the usefulness of biomarkers, since I am not an expert in this area, and they proved to be reliable, widely used indicators in research on stress. If there is any problem with them, the Authors gave a reasonable explanation referring to the specificity of local conditions. However, a lack of correlation between objective and subjective measures provokes the question of why it is that. Is there any answer to this question? I would like to know the Authors’ opinion on this matter. Perhaps, the significant correlation could occur at some level of experienced stress, but not at the other? The obtained PSS mean value (19.24 – line #346) suggests that level of stress in this sample should be considered as moderated (line #291). How it should be interpreted in comparison to non-pregnant females?
Also, I am a bit skeptical about the choice of the Perceived Stress Scale in this study. The fact that the PSS is the most commonly used self-report measure of subjective stress doesn’t give a sufficient reason to use it in this research. I agree with the Authors that potential stressors would have an impact on one’s well-being on the condition that they are perceived and interpreted as a source of discomfort. But some doubts may arise when we look at what this version of the PSS actually measures The Authors stated: It measures the degree to which situations are considered unpredictable, uncontrollable, and burdensome by the respondent during the past month (lines # 137 & 138). I would like to emphasize the fact that this scale was developed in the individualistic Western cultural context, where being in control / having control over things is extremely important for an individual, so losing control may be a huge source of distress. We know from cross-cultural psychology that the need to control may not be a universal one. For example, in collectivistic cultures, a tendency to have an external locus of control (believing that control is outside a person) is more pronounced than in individualistic cultures. Perhaps not being in control is not such a stressful condition for people, and it is more acceptable than in other cultures. So, being unable to control the important things (item #2), being able to control irritations (item #7), or having things outside control (item #9) might be irrelevant to Kenyan women’s lives. Stressors might come from somewhere else. I wonder whether the Authors would agree with my comments on the PSS as not being sensitive culturally. By the way, a reliability coefficient of the scale is not provided in the study.
The Authors decided to include intimate partner violence into the set of selected sociodemographic predictors. In the literature domestic violence is recognized as a risk factor of stress, especially when it happens during pregnancy. Regarding Kenyan women, we could find on the internet the following information:
Violence is a daily reality for women and girls across Kenya. According to government data, 45 percent of women and girls aged 15 to 49 have experienced physical violence and 14 percent have experienced sexual violence. Many cases are not reported to authorities and few women get justice or receive medical care. https://www.hrw.org/news/2020/04/08/tackling-kenyas-domestic-violence-amid-covid-19-crisis
Also, in Makayoto, L. A., Omolo, J., Kamweya, A. M., Harder, V. S., & Mutai, J. (2013). Prevalence and associated factors of intimate partner violence among pregnant women attending Kisumu District Hospital, Kenya. Maternal and Child Health Journal, 17(3), 441–447. https://doi.org/10.1007/s10995-012-1015-x :
One hundred and ten (37 %) of them experienced at least one form of IPV during pregnancy. Psychological violence was the most common (29 %), followed by sexual (12 %), and then physical (10 %).
In the light of the above figures, the low percentage of pregnant women who experienced IPV in the sample: 18,24% - is quite puzzling. Perhaps, it is because of how a question about being subjected to violence was put. I think the list of violent acts is incomplete. There is a lack of psychological violence listed, such as being insulted, belittled, humiliated, scared, threatened to hurt, etc.
Or, maybe such low numbers of women – victims of violence - are due to the specific features of the sample. In contrast to information about Migori County, provided by the Authors (lines 245 – 258), the participants of the study are very well educated: almost one-third of the sample attained a tertiary education as compared to 2% of the County population. Perhaps, the high education level of partners might work as a protective from domestic violence factor.
In conclusion, while I find the topic important, I think more should be done to clarify the points I raised in my comments.
Author Response
Response to Reviewer 2 Comments
Point 1: .....However, a lack of correlation between objective and subjective measures provokes the question of why it is that. Is there any answer to this question? I would like to know the Authors’ opinion on this matter.
Response 1: It is not surprising that we did not find a correlation between the psychological measure (PSS) and the biological measures (HCC and HCNC) and a number of studies have had similar results e.g. Braig et al 2016., Krammer et al 2009., Scharlau et al 2018., Musana J et al 2019. and Van der Voorn et al 2018. Variations in populations studied, methodological differences e.g. use of different stress measures, timing of obtaining the measures and the fact that the two measures might be giving information on different aspects of woman’s stress spectrum could have contributed to the lack of associations.
Point 2: The obtained PSS mean value (19.24 – line #346) suggests that level of stress in this sample should be considered as moderated (line #291). How it should be interpreted in comparison to non-pregnant females?
Response 2: A non-pregnant female sample could have provided an appropriate comparison group for the evaluations of hair steroids together with PSS in a population such as ours and this has to be considered in design of future studies. This was a recognized limitation of the parent cohort study and subsequently of this secondary data analysis.
Point 3: Choice of Perceived Stress Scale (PSS) for the study
Response 3: We appreciate the useful comments the reviewer has made regarding the use of PSS in our population. We acknowledge that the PSS used in our study did not undergo full psychometric testing in our study population and hence we were not able to give a reliability coefficient for our study population. It is possible that other psychological stress measures such as the pregnancy-specific anxiety measure might have given different results (Hoffman et al 2016). However, the PSS used in our study was translated into the local languages and underwent pretest for feasibility for cultural appropriateness through qualitative cognitive interviews with a focus group involving 10 pregnant women attending antenatal clinic at the study site hospital. Overall, the women reported the PSS questions were general enough to easily answer and the response options were simple to grasp and understand. Also, the results of the pretest indicated no need to modify the PSS questions. Additionally, PSS is the most widely used stress measure in varied cultural settings including in Sub-Saharan Africa and with very good internal consistency (alphas range of 0.69 to 0.91 across studies).
Point 4: Intimate partner violence.
Response 4: We thank the reviewer for the in-depth discussion on intimate partner violence and the additional references provided. We have incorporated these in our discussion (section 4.7, lines 494-510) and offered possible reasons of the reported low prevalence of IPV, and its nature of association with stress measures in this particular study population in contrast to what is reported in other literature.
Reviewer 3 Report
Dear authors,
The paper entitled "Maternal sociodemographic factors and antenatal stress" is fascinating. The authors have described most of the parts in great detail. There are small gaps to be covered in the manuscript. My comments for this paper are as follows:
- Line 66: Please delete the sentence "Available literature is reviewed below."
- (Line 204) "Table 1 below summarizes the currently available literature on selected sociodemographic determinants of ........ " Please add information about which papers showed those associations between exposure and outcome. You need to add the reference numbers in the table.
- Did you access the women's current history of antenatal depression? If the pregnant ladies have antenatal depression beforehand, they need to be excluded from the inclusion criteria. It should be added to your inclusion and exclusion criteria.
- Please add the reference after the sentence "Details on lab processing" (Line 275).
- Your study showed 18.24% of women reported intimate partner violence (IPV). IPV is a sensitive topic to ask during data collection. Please explain how it was reported in your methods section.
- It is confusing for readers to grab Table 4-7, A1-A5. Rather than writing p-value, confidence interval, and beta estimates, please change them to beta estimates, 95% CI, and p-value format. Also, it is understandable that CI has negative values. So rather than writing -0.14-0.14, I suggest changing the format to -0.14, 0.14. It will be less confusing to the readers.
- I suggest adding some evidence to the IPV during pregnancy and its impact on the infant's health in your subsection 4.7. You could use a South African birth cohort as a reference (Koen et al. 2014. Intimate partner violence: associations with low infant birth weight in a South African birth cohort. Metab Brain Dis).
- The self-reported answers can create bias in the findings. It should be discussed and added in the limitation.
Author Response
Response to Reviewer 3 Comments
Point 1: Line 66: Please delete the sentence "Available literature is reviewed below."
Response 1: Done - please see markup on revised manuscript.
Point 2: (Line 204) "Table 1 below summarizes the currently available literature on selected sociodemographic determinants of ........ " Please add information about which papers showed those associations between exposure and outcome. You need to add the reference numbers in the table.
Response 2: Done - please see revised Table 1 on revised manuscript.
Point 3: Did you access the women's current history of antenatal depression? If the pregnant ladies have antenatal depression beforehand, they need to be excluded from the inclusion criteria. It should be added to your inclusion and exclusion criteria.
Response 3: From the parent cohort study (Musana J et al 2019) women who had cognitive impairment or diagnosed mental health conditions were excluded from the study. This has been added. Please see line 269.
Point 4: Please add the reference after the sentence "Details on lab processing" (Line 275).
Response 4: Done.
Point 5: Your study showed 18.24% of women reported intimate partner violence (IPV). IPV is a sensitive topic to ask during data collection. Please explain how it was reported in your methods section.
Response 5: Please see markup under section 2.1, lines 280-283.
Point 6(a): It is confusing for readers to grab Table 4-7, A1-A5. Rather than writing p-value, confidence interval, and beta estimates, please change them to beta estimates, 95% CI, and p-value format.
Response 6(a): Upon suggestion of reviewer 1, Tables 4-7 have been summarized into Table 4 and detailed tables in the "beta estimates, 95% CI, and p-value" format suggested are under Appendix B. All tables have been modified to this format. Please see appendices A and B.
Point 6(b): Also, it is understandable that CI has negative values. So rather than writing -0.14-0.14, I suggest changing the format to -0.14, 0.14. It will be less confusing to the readers.
Response 6(b): Excellent suggestion. Done.
Point 7: I suggest adding some evidence to the IPV during pregnancy and its impact on the infant's health in your subsection 4.7. You could use a South African birth cohort as a reference (Koen et al. 2014. Intimate partner violence: associations with low infant birth weight in a South African birth cohort. Metab Brain Dis).
Response 7: Stronger evidence in the form of systematic reviews and meta-analyses is already presented in the last paragraph of subsection 4.7, lines 537-540. Nevertheless, thank you for the additional reference.
Point 8: The self-reported answers can create bias in the findings. It should be discussed and added in the limitation.
Response 8: Good recommendation. This had been discussed under section 1.2, lines 95-97, and now added to limitations, lines 554-556.